# Analyses of Public Attention and Sentiments towards Different COVID-19 Vaccines Using Data Mining Techniques

**DOI:** 10.3390/vaccines10050661

**Published:** 2022-04-22

**Authors:** Muhammad Faheem Mushtaq, Mian Muhammad Sadiq Fareed, Mubarak Almutairi, Saleem Ullah, Gulnaz Ahmed, Kashif Munir

**Affiliations:** 1Department of Computer Science, Khwaja Fareed University of Engineering and Information Technology, Rahim Yar Khan 64200, Pakistan; cosc202501010@kfueit.edu.pk (M.F.M.); saleem.ullah@kfueit.edu.pk (S.U.); gulnaz.ahmad@kfueit.edu.pk (G.A.); 2College of Computer Science and Engineering, University of Hafr Al Batin, Hafr Al Batin 31991, Saudi Arabia; 3Department of Information Technology, Faculty of Information Technology and Management, Khawaja Fareed University of Engineering and Information Technology, Rahim Yar Khan 64200, Pakistan; kashif.munir@kfueit.edu.pk

**Keywords:** COVID-19, VADER, vaccine, Moderna, Covaxin, Pfizer, Sinopharm, data mining

## Abstract

COVID-19 is a widely spread disease, and in order to overcome its spread, vaccination is necessary. Different vaccines are available in the market and people have different sentiments about different vaccines. This study aims to identify variations and explore temporal trends in the sentiments of tweets related to different COVID-19 vaccines (Covaxin, Moderna, Pfizer, and Sinopharm). We used the Valence Aware Dictionary and Sentiment Reasoner (VADER) tool to analyze the public sentiments related to each vaccine separately and identify whether the sentiments are positive (compound ≥ 0.05), negative (compound ≤ −0.05), or neutral (−0.05 < compound < 0.05). Then, we analyzed tweets related to each vaccine further to find the time trends and geographical distribution of sentiments in different regions. According to our data, overall sentiments about each vaccine are neutral. Covaxin is associated with 28% positive sentiments and Moderna with 37% positive sentiments. In the temporal analysis, we found that tweets related to each vaccine increased in different time frames. Pfizer- and Sinopharm-related tweets increased in August 2021, whereas tweets related to Covaxin increased in July 2021. Geographically, the highest sentiment score (0.9682) is for Covaxin from India, while Moderna has the highest sentiment score (0.9638) from the USA. Overall, this study shows that public sentiments about COVID-19 vaccines have changed over time and geographically. The sentiment analysis can give insights into time trends that can help policymakers to develop their policies according to the requirements and enhance vaccination programs.

## 1. Introduction

COVID-19 is a respiratory disease, caused by severe acute respiratory syndrome coronavirus 2 (SARS-CoV-2). COVID-19 can cause sore throat, shortness of breath, and respiratory illness. COVID-19’s first case was detected in December 2019 in Wuhan city, China [1]. It spread quickly, from person to person and country to country. Within a few days, it became a global disease that created a pandemic situation worldwide. The cases of COVID-19 reached above 100 million, including 2 million deaths by the end of January 2021 [2]. This pandemic has affected people’s lives worldwide. Many steps are being taken by world governments to control the spread of this pandemic disease and its effects. Although all the steps work effectively, vaccination is one of the most prominent steps taken by governments to reduce the effects of this viral disease.

Many vaccines have been introduced in markets across the world for emergency use since December 2020 [3]. Further, 50 more vaccines are under development [4]. On 31 December 2020, the WHO’s Emergency Use List (EUL) was updated with the Pfizer/BioNTech Comirnaty vaccine [5]. On 16 February, EUL was administered to the SII/Covishield and AstraZeneca/AZD1222 vaccines (designed by AstraZeneca/Oxford and manufactured by Serum Institute of India and SK Bio, respectively) [6]. Johnson & Johnson’s Janssen/Ad26.COV 2.S was approved for EUL on 12 March 2021 [7]. On 30 April 2021, EUL approved the Moderna COVID-19 vaccine (mRNA 1273), and the Sinopharm COVID-19 vaccine was approved on 7 May 2021 [8]. The Sinopharm vaccine was developed by the Beijing Bio-Institute of Biological Products, a subsidiary of China’s National Biotech Group [9].

To control the spread of the COVID-19 outbreak, a satisfactory proportion of the population must be vaccinated, which is almost 67% for COVID-19 [10]. Therefore, vaccination is necessary, although there are questions as to the extent to which these vaccines protect us from COVID-19 [11]. Different countries have adopted different mechanisms for the effective vaccination of their people; however, many people have expressed hesitancy towards vaccination. UK and European general population data show a mostly positive attitude towards vaccines; research also shows that there is still certainty or mistrust about the safety and effectiveness of vaccines in a substantial (10%) proportion of adults in the UK and European general population [12]. In Europe and the UK, 26% of adults were unwilling to receive a dose of a vaccine during the early pandemic (April 2020) [13]. Similarly, other studies show that one quarter of French [14] and US [15] adults are not ready for vaccination.

After COVID-19 spread around the world, many conspiracy theories appeared on different media platforms opposing vaccination against it. Many questions arose about vaccination quality, dose standards, religious buy-in, and suspicion about the presence of live viruses in vaccines [16]. People of different countries have shown different behavior towards vaccines due to misinformation and anti-vaccination sentiments. Misinformation and anti-vaccination sentiments reduce the effectiveness of vaccination. To understand people’s behavior during such situations, social media can play an important role [17]. People have expressed their concerns about COVID-19 using different social media platforms. Twitter is globally known as a popular social media platform, so we use Twitter as the data source in our study.

There is a need to analyze the public attitudes towards COVID-19 vaccines to better understand them, and this study sought to examine the public’s attitudes. This will help governments to reduce the hesitancy among individuals about COVID-19 vaccines and build their confidence in vaccines. The main contribution of this study is to analyze the sentiments of people regarding four vaccines (Covaxin, Moderna, Pfizer and, Sinopharm). The comparison of sentiment analysis of each vaccine shows different statistics in different time frames and different regions of the world. By using this proposed research study, policymakers can improve the efficiency of vaccination among populations where positive sentiments about the COVID-19 vaccine are low. The government and policymakers can compare locations in which positive sentiments are high in comparison with locations where positive sentiments are low and determine the factors that affect people’s health. Our temporal analysis will also help in identifying the time frame in which there is significant change occurring in public sentiments towards these vaccines.

## 2. Background

During the COVID-19 pandemic, different research and surveys have been conducted to overcome the effects of this disease. Vaccine hesitancy in a population is a basic hurdle in the control of vaccine-preventable illnesses. A long-term lockdown is not an efficient method in many developing countries owing to economic instability. The uptake of vaccinations may be the only method to restrict the pandemic’s endurance [18]. As vaccines play a great role in reducing the effects of COVID-19, researchers have collected data from different online sources to analyze public sentiments about different COVID-19 vaccines. Twitter is one of the most popular social media platforms used to research health issues [19]. Twitter shows real-time public attention, behavior, and attitudes in different locations [20]. There are 100 million active users present on Twitter [21] and almost 500 million tweets posted every day [22].

In previous studies, tweets have been used successfully to analyze public sentiments about vaccines other than COVID-19. Raghupathi et al. used the frequency-inverse document frequency technique to analyze public sentiments towards vaccination, and their findings show that most people have concerns about newly developed vaccines [23]. Due et al. constructed a model of tweets related to the HPV vaccine, and the model shows that public sentiments changed in early 2017 and the US varied significantly [24]. Similarly, a few studies have been performed to analyze public sentiments through COVID-19-related tweets. For example, Dubey et al. analyzed COVID-19 vaccine-related tweet sentiments from 14 January 2021 to 18 January 2021, which were posted in India only [25]. Paul et al. analyzed public attitudes towards vaccines using ordinary least squares regression and multinomial regression methods. This survey showed that 16% of participants had a high level of mistrust towards vaccines. Overall, 14% of participants showed unwillingness to receive the COVID-19 vaccine [26]. Kreps et al. used the conjoint analysis method to determine the role of vaccine attributes towards vaccination. This survey shows that different attributes of a vaccine affect people’s willingness to receive vaccinations. An attribute that had a larger effect on individual vaccine preference was efficacy [27]. Doaa et al. performed a cross-sectional survey to analyze intention and attitudes about the COVID-19 vaccine in Egyptian adults from 7 January 2021 to 30 March 2021. Among 1011 Egyptians, 54% showed hesitancy towards the COVID-19 vaccine, 21% of them showed non-acceptance of the vaccine, and 25% among them showed a willingness to receive the vaccine. Approximately 27.1%, 6.9%, and 4.5% of respondents preferred the Pfizer, Chinese, and AstraZeneca vaccines, respectively [28]. Similarly, Al-Qerem et al. studied the attitudes of Jordanian young adults towards the COVID-19 vaccine using a non-parametric statistical test. A total of 1897 participants took this survey and 19.9% of them were ready to receive a dose of the COVID-19 vaccine. Participants showed differences in the acceptance of different COVID-19 vaccines, and specific knowledge about vaccines was a significant predictor of vaccine acceptance [29]. Khan et al. performed a mixed-method survey to analyze the attitudes of the Pakistani population towards the COVID-19 vaccine from 15 September 2020 to 30 November 2020. The total number of participants included in this survey was 1003, and 71.29% among them reported that they will be vaccinated when the vaccine is made available [30].

Vaccines have helped significantly to overcome the effects of the COVID-19 pandemic. A global survey of participants from 19 countries showed that acceptance rates ranged between less than 55% to over 90% and the overall result showed that the acceptance rate was 71.5% [31]. The vaccination rate in the United Arab Emirates and Bahrain in January 2021 was high as compared to other countries. The United Arab Emirates and Bahrain vaccination rates were 33 and 11.56 per 100 people, respectively, whereas in other countries, it was less than 2 per 100 people [32]. In Jordan, it was less than 0.5% [33]. These rates in the Middle East were reported before vaccine availability; therefore, the situation may be different now. As discussed earlier, different vaccines evoke different sentiments in people; therefore, our study aims to investigate these sentiments and find associations, if any, between the different vaccines.

## 3. Materials and Methods

### 3.1. Data Collection

We used different combinations of hashtags and keywords ‘‘(#COVID OR COVID-19 OR #COVID19) AND (#VACCINE OR VACCINES ORVACCINATION OR VACINE OR)” to explore tweets related to COVID-19 vaccination between 12 December 2020 and 24 October 2021. We used the Twitter social media platform to collect data as it offers an API that enables developers to integrate Twitter with other applications [34]. We employed the tweepy Python package to collect the tweet-related data and then used a filter to exclude retweets [35]. Our dataset contained tweets related to Pfizer/BioNTech, Sinopharm, Sinovac, Moderna, Oxford/AstraZeneca, Covaxin, Sputnik V., and other COVID-19-related tweets.

### 3.2. Data Statistics

All vaccine tweets were separated from the dataset for data mining based on their names, and the total number of tweets was 212,982. The dataset contained 63,545 and 40,552 tweets about Covaxin and Moderna, respectively. The majority of tweets were related to Covaxin and Moderna, which constituted around 44% of all vaccine-related tweets. Similarly, the percentage of tweets about Pfizer and Sinopharm was 7.4% and 4.6%, respectively. Overall, 70% of tweets contained user locations. The percentages of tweets related to the specific vaccines are shown in Table 1.

### 3.3. Pre-Processing

Pre-processing is an important step in a study related to data science. It uses several Natural Language Processing (NLP) techniques to process the data according to requirements. After the collection of data, these techniques are used for cleaning data to obtain better results. NLP techniques used for pre-processing were the conversion of tweet text to lowercase and removal of punctuation, Uniform Resource Locators (URLs), and stop words from tweets. In the end, the stemming and lemmatization technique was used for further pre-processing. The stemming and lemmatization technique is used to convert a word into its root word (e.g., caring into care) but the difference between these techniques is that lemmatization is more intelligent than stemming [34]. For example, if we wish to convert the word better into its root word, then stemming may merely chop it into “bett” or “bet” while lemmatization will convert into “good”.

### 3.4. Sentiment Analysis

Sentiment analysis is used to analyze people’s attitudes and behavior. It is a significant task in NLP that uses rule-based, supervised, or unsupervised machine learning techniques to classify people’s sentiments into multiple categories [36]. One of the most common classifications of sentiments is to divide them into positive, negative, and neutral categories. In this research, the Valence Aware Dictionary and Sentiment Reasoner (VADER) [37] tool was used for sentiment analysis. The Textblob [37] library is also used for sentiment analysis, but VADER is more valid than TextBlob as it was developed to analyze sentiments, especially for social media websites. Its F1 score (0.96) is higher than that of any other machine learning model; it is even more accurate than any human raters (F1 = 0.84) [38]. It analyzes each text document, including emojis, and generates a normalized compound score ranging from −1 (extremely negative) to +1 (extremely positive). Normally, sentiments are classified into three categories: positive, negative, and neutral. This study classified sentiments into positive (compound score ≥ 0.05) and negative (compound score ≤ −0.05) categories based on the compound score. For the neutral category, the compound score must be greater than −0.05 and less than 0.05.

The TextBlob sentiment analyzer returns the result of input in two categories. One is the polarity, which is a float number. It lies within a range from −1 to +1. A score close to +1 is considered positive and a score close to −1 is considered negative. The second is subjectivity, which is also a float number that lies between 0 and 1.

The subjectivity score represents whether a statement is more opinion- or fact-based. VADER returns output with positive, negative, neutral, and compound classification. The compound score is the sum of the positive, negative, and neutral scores. The compound score is calculated by adding together the sentiment scores of all the words in a text, adjusting it according to VADER rules, and then normalizing the score between +1 (most extreme positive) to −1 (most extreme negative). The following equation is used to normalize the compound score:(1)$$x=\frac{x}{\sqrt{x^{2}+\alpha }}$$
where *x* is the sum of the valence score words in a text and α = normalization constant (default value = 15). TextBlob only analyzes the known words, and it ignores unfamiliar words [37]. It considers only subjectivity and polarity, whereas VADER gives output with three classifications, as well as a compound score [39]. VADER is specifically developed for social media data such as Twitter and Facebook, so it gives good results related to sentiments. The total numbers of tweets with positive, negative, and neutral sentiments using TextBlob and VADER are given in Table 2.

We used the McNemar [40] test to compare the results obtained using TextBlob and VADER. We found that sentiments obtained using TextBlob and VADER were statistically significantly different. We obtained a *p*-value less than 0.05; therefore, the null hypothesis was rejected in the McNemar test.

After finding sentiment results using VADER, the word cloud technique was used for the representation of each vaccine sentiment. Word cloud shows the importance of text by using different font colors and sizes [41]. Word cloud [41] and matplotlib [42] libraries are used with Python to create a word cloud of these sentiments. All these functions are executed in Jupyter Notebook because it is a great tool for scientists to share their code, related computation, and documentation [43].

### 3.5. Temporal Analysis

After using a sentiment analyzer and obtaining the sentiment score of each tweet, we obtained the daily average of sentiment scores and plotted the distribution over time. We compared the sentiment scores of vaccines individually and also analyzed the public sentiments on different dates to check for significant changes in public sentiments.

### 3.6. Geographical Analysis

We used a heat map to map the sentiments based on user location. In this study, we analyzed tweets geographically and, after this, all the tweets were separated based on user location. Then, the top 5 countries with the maximum number of tweets based on user location were visualized to assess the sentiments geographically. We used a bar chart to visually analyze the tweets geographically.

## 4. Results and Discussion

### 4.1. Data Separation

We collected a total of 212,982 tweets related to COVID-19 vaccines in our dataset. Specific vaccine tweets were separated based on their names. We separated our data into four vaccines (Covaxin, Moderna, Pfizer, and Sinopharm). Statistics of a specific vaccine are mentioned in Table 1. Most of the tweets contained the user location for geographical analysis.

### 4.2. Sentiment Analysis

In this study, VADER was used for the sentiment analysis of all COVID-19 vaccine-related tweets. We used the following values to calculate the sentiment score: compound score ≥ 0.05 for positive, compound score ≤ −0.05 for negative, and −0.05 < compound score < 0.05 for neutral sentiments. After applying VADER, we obtained the following results.

Figure 1a represents the sentiments of people about the COVID-19 Covaxin vaccine. Most of the sentiments are neutral. In total, 28% of tweets about Covaxin are positive and only 8% are negative. Figure 1b represents the sentiments of people about the COVID-19 Moderna vaccine. People’s sentiments about Moderna are 46% neutral, 37% positive, and 17% negative. Similarly, Figure 1c,d represent the sentiments about Pfizer and Sinopharm, respectively. People’s sentiments about the Pfizer vaccine are 45% neutral, 38% positive, and 17% negative. Tweets related to Sinopharm are 52% neutral, 28% positive, and 20% are negative. The vaccines elicited different feelings among people, as Raghupathi et al. demonstrate in their study, where people were concerned about receiving newly developed vaccines [23]. These statistics show that people’s sentiments about Covaxin are more positive as compared to Moderna, Pfizer, and Sinopharm. Moderna ranks second in terms of positive sentiments. The sentiments related to each vaccine are mostly neutral and positive, and in another study, Paul et al. found that only 14% of respondents had negative attitudes towards vaccines [26].

The words used most frequently in the tweets are listed in Table 3. Effective, dose, got, first, and vaccinated are the main words related to positive sentiments, and death, report, victims, and pain are related to negative sentiments. As many rumors and misinformation is circulating, especially on social media, people wish to know about the efficiency of vaccines. They wish to know about the quality of vaccines and dose standards [30].

As Pfizer [44] and Moderna [45] both are developed by American companies and both completed their trials one after another, people discussed these vaccines at the same time in their tweets. This is why the word cloud for Moderna contains the word “Pfizer” and the word cloud for the Pfizer vaccine contains “Moderna” with a high frequency. The most frequent words in Sinopharm tweets are Sinopharm, vaccine, man, good, day, health, receiving, and vaccinated as depicted in Figure 2. Most words occurring in Sinopharm-related tweets are slightly different as compared to the other vaccines.

### 4.3. Temporal Analysis

#### 4.3.1. Covaxin

Tweets were collected from 12 December 2020 to 24 October 2021, and the total number of days during this duration was 317. The maximum number of tweets related to Covaxin tweeted in a day was 1996 (19 July 2021). The average score for Covaxin-related tweets was 262.58 per day, with an overall mean of 0.10. In this study, we found a significant change in the number of tweets per day after 6 June 2021, and in July, it reached its maximum number. Then, the number of tweets dropped to 500 and below in August onwards, with the exception of one day (12 October 2021).

#### 4.3.2. Moderna

The average score of tweets related to the Moderna COVID-19 vaccine was 139.35 per day. The maximum number of Moderna vaccine tweets in a single day was 588 (16 April 2021). In April, the number of tweets changed significantly from 4 April 2021 to 21 April 2021. Then, it was seen to fluctuate as before.

#### 4.3.3. Pfizer

The average score for Pfizer tweets was 63.0 per day. Our dataset contained 18,396 tweets related to the Pfizer COVID-19 vaccine. The maximum number of tweets related to Pfizer in a single day was 245, which was found on 23 August 2021. We observed that the total number of tweets in a single day was below 100 from 12 December 2021 to 28 March 2021. It was above one hundred only on the 8th and 9th of January. The total number of tweets in a single day reached above 150 on 24 May 2021, and in August, it reached 245. In August 2021, people showed more concern about the Pfizer vaccine in their tweets, according to our data.

#### 4.3.4. Sinopharm

There were a total of 8901 tweets related to Sinopharm present in our dataset. The average number of tweets was 33.79 and the maximum numbers of tweets in a single day was 856 (13 August 2021). According to our data, the total number of tweets related to Sinopharm in a single day was below 100. These numbers increased in March (20 March 2021) and August (13 August 2021) only and reached above one hundred.

Figure 3 shows the sentiment distribution over time, and the red line represents the moving average of fourteen days. We also calculated the proportion of tweets for every type of sentiment (See Figure 4). We plotted the frequency of all four vaccines’ positive, negative, and neutral sentiments on a graph from 6 March 2021 to 20 March 2021. During this duration, the number of positive, negative, and neutral tweets related to Covaxin was 477, 172, and 452, respectively. The number of positive, negative, and neutral tweets related to Moderna was 675, 301, and 855, respectively. Similarly, for Pfizer, the number of positive, negative, and neutral tweets was 290, 128, and 372, respectively. Further, for Sinopharm, 186, 60, and 321 were the frequency of positive, negative, and neutral tweets, respectively. These statistics show that Moderna’s positivity rate was high from 6 March 2021 to 20 March 2021 as compared to the other vaccines. The total number of Moderna tweets was also high during this period.

### 4.4. Geographical Analysis

Our dataset contained 212,982 tweets, and this included 150,242 with the user location. We analyzed 130,602 user locations of tweets because these tweets mentioned Covaxin, Moderna, Pfizer, and Sinopharm.

#### 4.4.1. Covaxin

Out of 63,545 tweets related to the Covaxin vaccine, 34,662 tweets had user locations. Our data show that most of the users who tweeted about the Covaxin COVID-19 vaccine were from India.

#### 4.4.2. Moderna

The Moderna vaccine has been discussed in 40,552 tweets and 31,500 of them mention the location. According to this study, most of the users who tweeted about the Moderna COVID-19 vaccine were from the United States of America.

#### 4.4.3. Pfizer

Pfizer vaccine-related tweets had a location in 15,721 tweets out of 18,396. Our data show that most of the users who tweeted about the Covaxin COVID-19 vaccine were from India.

#### 4.4.4. Sinopharm

Out of 8109 tweets related to the Sinopharm vaccine, 6882 tweets had user locations. Our data show that most of the users who tweeted about the Covaxin COVID-19 vaccine were from China.

Both the Covaxin and Pfizer vaccines had a large number of tweets from India. We analyzed the data of five countries (India, USA, Canada, UK, and China). We separated the user locations based on the states and capitals of these countries.

Our study found that most of the tweets related to the four vaccines were from India. Tweets related to the Covaxin vaccine had large numbers of users (24,411) from India as compared to other countries. Moderna had the most users from the USA, whereas tweets related to Sinopharm had the most users from China.

Figure 5 shows the heat map of the sentiment scores of the countries. The maximum sentiment score of tweets related to the Covaxin vaccine posted from India was 0.97 and it was the highest among the selected five countries. In the USA, the highest sentiment score of tweets related to the Covaxin vaccine was 0.88. On the other hand, Canada had the lowest sentiment score of −0.66 for tweets related to the Covaxin vaccine. The highest sentiment score (0.96) of tweets related to Moderna was found in the USA, and the lowest sentiment score was also seen in the USA, which was also −0.96. India ranked second for the lowest sentiment score (−0.95) of tweets related to the Moderna vaccine. Tweets related to the Pfizer vaccine had the highest sentiment score of 0.9231 and the lowest sentiment score of −0.9217 from the UK. The second highest sentiment score of tweets related to the Pfizer vaccine was from the USA and it was 0.9153. China had the highest sentiment score of tweets related to Sinopharm, which was 0.8074. India had the lowest sentiment score of tweets related to Sinopharm, and it was −0.75.

The average mean of the sentiments related to the four COVID-19 vaccine is shown in Table 4, and the vaccine coverage in the mentioned five countries (India, USA, Canada, UK, and China) is shown in Figure 6. Table 4 shows that the average sentiment score of Covaxin was 0.0046, with 24,313 tweets in India. The USA had the lowest average sentiment score (−0.0007), with 4632 tweets related to Moderna. The Moderna vaccine had the highest average sentiment score of 0.0551, with 71 tweets in China. The Analysis of Variance (ANOVA) [46] and Tukey (CI = 95%) [47] methods were used to examine differences in the sentiments of these five countries. These methods showed that India and China, the USA and China, Canada and China, and the UK and China had differences of 0.2061, 0.1816, 0.0805, and 0.2031 in sentiments related to the Covaxin vaccine, respectively. In the case of the Moderna vaccine, a large statistical difference of 0.0923 in sentiments was present between India and China. There was a large statistical difference of 0.0356 present between the USA and Canada in the case of the Pfizer vaccine. The statistical difference (0.1422) in the case of the Sinopharm vaccine was large between India and Canada.

## 5. Conclusions

In this study, we performed a sentiment analysis of 212,982 tweets related to the Covaxin, Moderna, Pfizer, and Sinopharm COVID-19 vaccines and found that their sentiments were mostly neutral. Covaxin, Moderna, Pfizer, and Sinopharm had positive sentiments of 28%, 37%, 38%, and 28%, respectively, while their negative sentiments totaled 8%, 17%, 17%, and 20%. These statistics show that Covaxin’s positivity rate is high, with 17,793 tweets, as compared to other vaccines. The number of tweets related to Covaxin increased significantly in October 2021 as it was due to be licensed for emergency use by WHO, whereas the number of tweets related to Moderna, Pfizer, and Sinopharm increased in June, August, and March, respectively. Public sentiments also changed geographically. Government officials and health policymakers should adopt effective vaccine education programs on the basis of timely changes and region-based sentiments. Government should track the changes in the sentiments of the public over time and try to detect the topics discussed by them. This can help them to strength the vaccination program and to achieve herd immunity among the population.

## 6. Limitations

Out of 212,982 tweets, our dataset contained only 130,602 tweets related to COVID-19 vaccines (Covaxin, Moderna, Pfizer, and Sinopharm) with a user location, and in the future, researchers should work on data with an increased number of tweets with user locations to find the rate of positive and negative sentiments in different locations. Secondly, after analyzing the tweet content, we found that VADER identified the sentiments of text correctly but it could not distinguish whether the COVID-19 vaccine was the subject of the statements. For example, a tweet with text “COVID is gone” expresses the confidence of the user but VADER considered it as neutral text.

## Figures and Tables

**Figure 1 vaccines-10-00661-f001:**
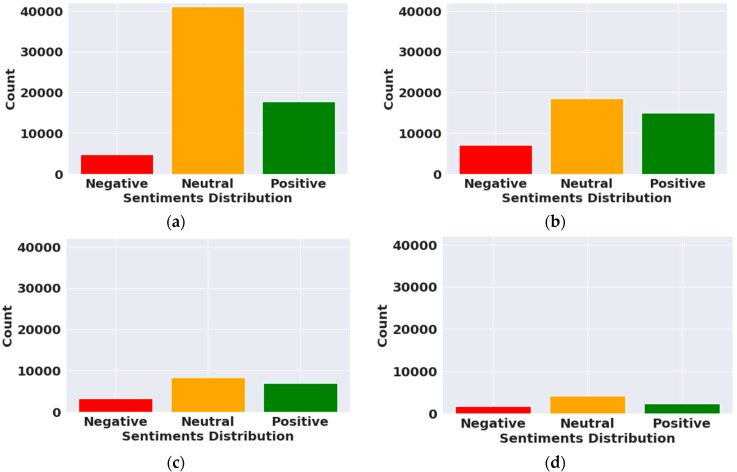
Distribution of COVID-19 vaccine sentiments: (**a**) Sentiment distribution of Covaxin; (**b**) sentiment distribution of Moderna; (**c**) sentiment distribution of Pfizer; (**d**) sentiment distribution of Sinopharm.

**Figure 2 vaccines-10-00661-f002:**
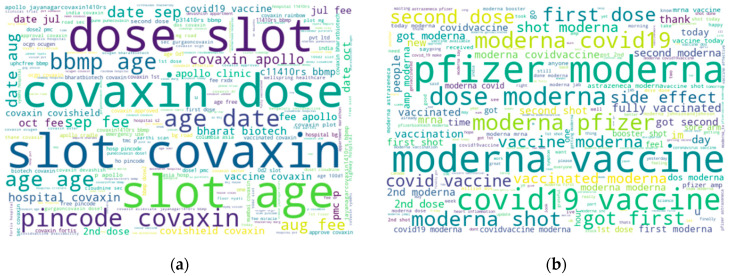
Word cloud of four different COVID-19 vaccines: (**a**) word cloud of Covaxin-related tweets; (**b**) word cloud of Moderna-related tweets; (**c**) word cloud of Pfizer-related tweets; (**d**) word cloud of Sinopharm-related tweets.

**Figure 3 vaccines-10-00661-f003:**
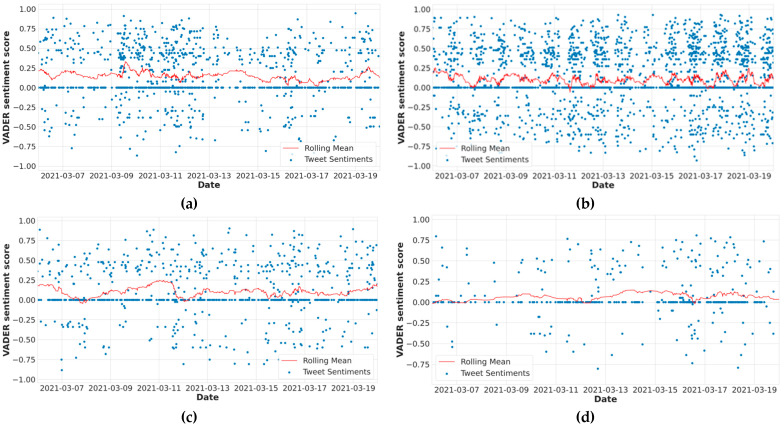
Different COVID-19 vaccines’ sentiment distribution over time and rolling mean: (**a**) Covaxin sentiment distribution over time and rolling mean; (**b**) Moderna sentiment distribution over time and rolling mean; (**c**) Pfizer sentiment distribution over time and rolling mean; (**d**) Sinopharm sentiment distribution over time and rolling mean.

**Figure 4 vaccines-10-00661-f004:**
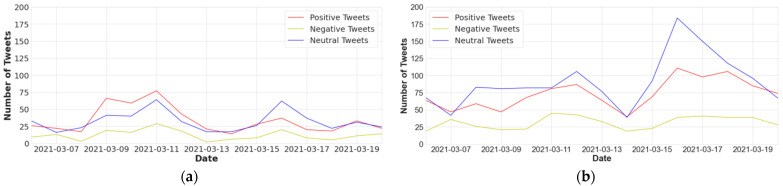
Distribution of four COVID-19 vaccines’ tweets across sentiment types: (**a**) distribution of Covaxin tweets across sentiment types; (**b**) distribution of Moderna tweets across sentiment types; (**c**) distribution of Pfizer tweets across sentiment types; (**d**) distribution of Sinopharm tweets across sentiment types.

**Figure 5 vaccines-10-00661-f005:**
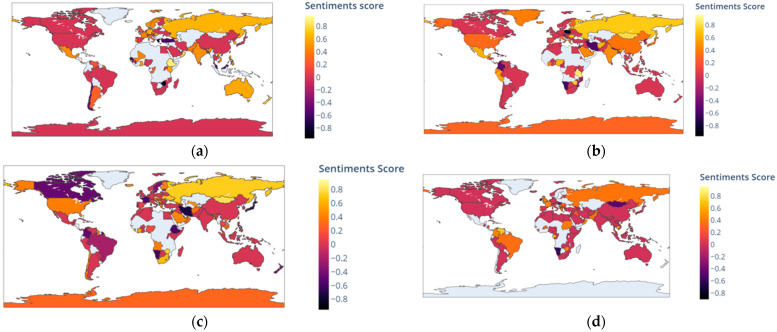
Heat map of four COVID-19 vaccines’ sentiment scores by country: (**a**) heat map of Covaxin; (**b**) heat map of Moderna; (**c**) heat map of Pfizer; (**d**) heat map of Sinopharm.

**Figure 6 vaccines-10-00661-f006:**
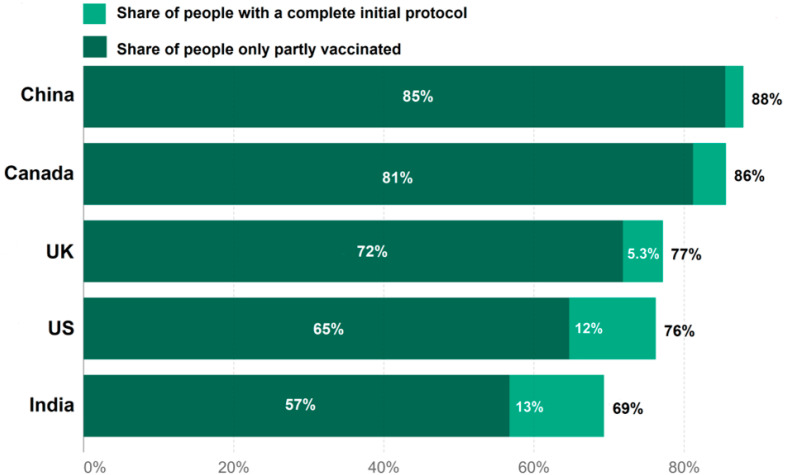
COVID-19 vaccine coverage in five countries.

**Table 1 vaccines-10-00661-t001:** The total number of tweets collected related to different COVID-19 vaccines.

Name	Number of Tweets	Percentage of Tweets
Covaxin	63,545	29.8%
Moderna	40,552	19%
Pfizer	18,396	8.6%
Sinopharm	8109	3.8%

**Table 2 vaccines-10-00661-t002:** The sentiment classification of tweets using TextBlob and VADER.

Sentiments	Number of Tweets Using TextBlob	Percentage of Tweets Using TextBlob	Number of Tweets Using VADER	Percentage of Tweets Using VADER
Positive	70,607	33%	77,573	36%
Negative	21,557	10%	32,085	15%
Neutral	120,818	57%	103,324	49%
Total	212,982	100%	212,982	100%

**Table 3 vaccines-10-00661-t003:** Most frequent words in COVID-19 vaccine-related tweets across sentiment types.

Sentiments	Frequent Words	
	Positive Sentiments	Negative Sentiments
Covaxin	Covaxin, age, slot, vaccine, dose, Bharat biotech, approval, free, India, covishield	Covaxin, hospital, vaccine, block, age, slot, emergency, dose, use, India
Moderna	Moderna, vaccine, Pfizer, COVID19, shot, got, dose, first, today, vaccinated	Moderna, vaccine, Pfizer, arm, sore, effect, hour, pain, report, death, Japan
Pfizer	Pfizer, vaccine, Moderna, Pfizerbiontech, COVID19, dose, effective, first, get, shot	Pfizer, Moderna, vaccine, Pfizerbiontech, covid19, AstraZeneca, death, report, people, victims
Sinopharm	Sinopharm, vaccine, China, COVID19, approved, Sinovac, Chinese, dose, use, got, vaccinated, effective	man, Sinopharm, vaccine, day, vaccinated, died, receiving, health, Sinovac, China

**Table 4 vaccines-10-00661-t004:** The average of sentiments and number of tweets in the top five most posted countries.

Country	Vaccine							
	Covaxin		Moderna		Pfizer		Sinopharm	
	Mean	Tweets	Moderna	Tweets	Pfizer	Tweets	Mean	Tweets
India	0.0046	24,313	−0.0372	1228	−0.0053	1028	0.0237	198
USA	0.0291	869	−0.0007	4632	0.0045	1188	−0.0164	53
Canada	0.1302	399	0.0104	2489	0.0401	1068	0.1659	34
UK	0.0076	392	0.0492	898	0.0007	740	0.0269	71
China	0.2107	09	0.0551	71	−0.0530	38	0.0912	818

## Data Availability

The supporting data for the findings of this study are available from the corresponding author on reasonable request.

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
