# Peer review of "Analyses of Public Attention and Sentiments towards Different COVID-19 Vaccines Using Data Mining Techniques"

_vaccines, 2022, doi:10.3390/vaccines10050661_

Round 1

Reviewer 1 Report

  • The subject of the research article is interesting to healthcare workers and virology researchers.
  • Overall, the article is properly written and represented. No English neither grammatical errors are noticed. Only one suggestion is to check the position of Table 3 since it intersects two long paragraphs.
  • The cited references are relevant to the topic and their numbers suffice the work.
  • The study design and approaches, data analysis and curation, and scientific written material are highly qualified and properly executed.
  • I strongly recommend publishing this piece of work.
  • Final Decision: Acceptance.

Reviewer 2 Report

Thanks to the authors. 

I may suggest to combine the Introduction and the Review of Literature into one section.

Delete the lines 86-89

Update the figures of the vaccine coverage in the mentioned countries.

Please explain why you chose four vaccines among others that had EUL

Also, explain why you concentrate on five countries to assess the spatial distribution. I believe it is important to consider the population number in these countries that will definitely affect this distribution. 

It is important to add a sub-section on the limitations of the study where all the mentioned gaps and others  can be added.

The conclusion section should be revised to avoid repeating the results and identify how these results can be used to support policymakers to have informed decisions and make a public health impact.

Author Response

Please find it in the attachment.

Reviewer 3 Report

This is a novel approach to measure feelings about various vaccines in social networks, the technique used is innovative in the field of vaccines, in addition to the interest in the topic itself, it is of interest to serve as a model for the use of this technique.

It would be better if section 2 were called Background because the authors have not really conducted a systematic review.

Data collection. 3.1.

The authors must consider that the readers of the journal vaccines may not be familiar with ICT. It should be explained in more detail how the database was created, and from where the data were extracted.   After careful reading of the paragraph, it seems to be deduced that the data came from twitter, but this is not specifically stated, it could also be derived from Facebook, Google searches, etc.

The methodology should be explained in detail so that a non-computer literate reader can understand it.

We employed the tweepy Python's package. The tweepy program reference should be provided.

Explain what stemming and lemmatization is and provide references.

Reference is made to Valence Aware Dictionary and Sentiment Reasoner (VADER), but no bibliographic reference is provided.

Similarly, Text blob library is mentioned but no bibliographic references are provided.

When comparing text blob and VADER some reference should be provided.

The data in table 2 are interesting.  But a column with percentages over the total number of messages analyzed should be added.

A statistical analysis should be performed in a contingency table comparing the rating with text blob and VADER.  The McNemar repeated measures statistic should be used.

The Kappa statistic should also be used to determine the degree of agreement. The calculations can be easily done with the openepi program freely available at www.openepi.com.

They should indicate with which programs they have made the various graphs Word cloud, maps, evolutions etc.

I do not understand the title of table 4, it is indicated that it is the average, but of what? The average per capita? The daily average?  Further down at the bottom of the table it is indicated that it refers to feelings. The authors should change the title of the table.

The authors indicate that in Table 4, there are no differences between countries.  The data should be presented as mean and standard deviation. They should perform a statistical test such as ANOVA or Kruskall Wallis to be able to say whether there are differences between countries. If differences are detected, they should perform a posteriori contrast by Scheffe's or Tukey's method to detect between which groups there are differences.

There is a section on results and discussion.  In the discussion the bibliography should be introduced, and the results interpreted in the light of the existing bibliography.

It is an interesting and innovative work, and I would like to see it published, but the authors need to work on it more.

Author Response

Response to Reviewer 3 Comments

The author would like to express their sincere appreciation for the reviewer’s insightful revisions and constructive suggestions. A significant improvement in the quality of the manuscript was achieved based on the suggestions and recommendations given. We have answered all the questions and the detail is given below point by point:

Point 1: It would be better if section 2 were called Background because the authors have not really conducted a systematic review.

Response 1: In the revised manuscript, Section 2 is renamed Background. The screenshot is attached below.

Point 2: The authors must consider that the readers of the journal vaccines may not be familiar with ICT. It should be explained in more detail how the database was created, and from where the data were extracted.   After careful reading of the paragraph, it seems to be deduced that the data came from twitter, but this is not specifically stated, it could also be derived from Facebook, Google searches, etc.

The methodology should be explained in detail so that a non-computer literate reader can understand it.

We employed the tweepy Python's package. The tweepy program reference should be provided.

Response 2: The revised manuscript adds references for the data collection methods used and the platform used to collect data on lines 145 to 148.  Referencing added at lines 146 and 148 with 34 and 35 numbers, respectively. The revised data collection section is written as:

“We used different combinations of hashtags and keywords ‘‘(#COVID OR COVID-19 OR #COVID19) AND (#VACCINE OR VACCINES ORVACCINATION OR VACCINE OR)”) to explore the tweets related to COVID-19 vaccination between December 12, 2020, to October 24, 2021. We used Twitter social media platform to collect data as it offers an API that enables developers to integrate Twitter with other applications [34]. We employed the tweepy Python package to collect the tweets-related data and then used a filter to exclude retweets [35]. Our dataset contains Pfizer/BioNTech, Sinopharm, Sinovac, Moderna, Oxford/AstraZeneca, Covaxin, Sputnik V., and other COVID-19 related tweets.”

Point 3: Explain what stemming and lemmatization is and provide references.

Response 3: In the revised manuscript, an explanation of stemming and lemmatization is added at line 167 to line 171 with reference number 34. This explanation is written as:

“The stemming and lemmatization technique is used to convert a word into its root word like (caring into care) but the difference between these techniques is that lemmatization is more intelligent than stemming [34]. For example, we want to convert the word better into its root word then stemming may just chop it into “bett” or “bet” but lemmatization will convert into “good”.”

Point 4: Reference is made to Valence Aware Dictionary and Sentiment Reasoner (VADER), but no bibliographic reference is provided. Similarly, Text blob library is mentioned but no bibliographic references are provided.

Response 4: In the revised version of the manuscript, references and bibliography are added to VADER and Textblob at lines 178 and 181. Now text look like as:

“In this research, Valence Aware Dictionary and Sentiment Reasoner (VADER) [37] tool is used for sentiment analysis. Textblob [37] library is also used for sentiment analysis but VADER is more valid than text blob as it is developed to analyze sentiments especially for social media websites. Its F1 score (0.96) is higher than any other machine learning model even it is more accurate than any human raters (F1 = 0.84) [38]”

Point 5: When comparing text blob and VADER some reference should be provided.

Response 5: In the revised version of the manuscript, references are added to lines 202 and 203 when comparing VADER and Textblob. The revised comparison is written as:

“Textblob only analyzes the known words, it ignores the unfamiliar words [37]. It considers only subjectivity and polarity whereas VADER gives output with three classifications as well as compound score [39]. VADER is specifically developed for social media data such as Twitter and Facebook so it gives us good results related to sentiments.”

Point 6: The data in table 2 are interesting.  But a column with percentages over the total number of messages analyzed should be added.

Response 6: In the revised version of the manuscript, percentage of sentiments using Textblob and VADEDR is added at line 207 and screenshot is given.

Point 7: A statistical analysis should be performed in a contingency table comparing the rating with text blob and VADER.  The McNemar repeated measures statistic should be used.

Response 7:. In the revised manuscript, the McNemar is used to detect differences between textblob and VADER results, and it is described on lines 208 to 211 as follows:

“We used the McNemar [40] test to compare the results obtained using textblob and VADER. We find that sentiments obtained using textblob and VADER are statistically significantly different. We obtained a p-value less than 0.05 therefore it rejects the null hypothesis in McNemar test.”

Point 8: The Kappa statistic should also be used to determine the degree of agreement. The calculations can be easily done with the openepi program freely available at www.openepi.com.

Response 8:.In the future, experiments we will add the Kappa statistics to determine the degree of agreement.

Point 9: They should indicate with which programs they have made the various graphs Word cloud, maps, evolutions etc.

Response 9:.In the revised manuscript, program used for plotting graphs and word cloud are mentioned on line 212 to 217 and it is written as:

“After finding sentiment results using VADER, the Word cloud technique is used for the representation of each vaccine sentiment. Word cloud shows the importance of text by using different font colors and sizes [40]. Word cloud [40] and matplotlib [41] libraries are used with Python to create a word cloud of these sentiments. All these functions are executed in Jupyter Notebook because it is a great tool for scientists to share their code, related computation, and documentation [42]. ”

Point 10: I do not understand the title of table 4, it is indicated that it is the average, but of what? The average per capita? The daily average?  Further down at the bottom of the table it is indicated that it refers to feelings. The authors should change the title of the table.

Response 10: The title of Table 4 is changed in the revised manuscript from “The average mean of tweets in our collected data related to COVID-19 vaccines for different countries of the World.”  to “The average of sentiments and number of tweets in the top five most posted countries

Point 11: The authors indicate that in Table 4, there are no differences between countries.  The data should be presented as mean and standard deviation. They should perform a statistical test such as ANOVA or Kruskall Wallis to be able to say whether there are differences between countries. If differences are detected, they should perform a posteriori contrast by Scheffe's or Tukey's method to detect between which groups there are differences.

Response 11: In the revised manuscript, we apply both ANOVA and Tukey methods to find statistical differences between countries and it is written as follows:

“The average mean of four COVID-19 vaccine sentiments is shown in Table 4. Table 4 shows that the average sentiment score of Covaxin is 0.0046 with 24313 numbers of tweets in India. The USA has the lowest average sentiment score (-0.0007) with 4632 tweets related to Moderna. The Moderna vaccine has the highest average sentiment score of 0.0551 with 71 numbers of tweets in China. The ANOVA (Analysis of Variance) [46] and Tukey(CI=95%) [47] methods are used to find difference in sentiments of these five countries. These methods show that India and China, USA and China, Canada and China and, UK and China have 0.2061, 0.1816, 0.0805, 0.2031of difference in sentiments related to Covaxin vaccine, respectively. In the case of the Moderna vaccine, a large statistical difference of 0.0923 in sentiments is present between India and China. There is a large statistical difference of 0.0356 is present between USA and Canada in the case of the Pfizer vaccine. The statistical difference (0.1422) in the case of the Sinopharm vaccine is large between India and Canada.”

Point 12: There is a section on results and discussion.  In the discussion the bibliography should be introduced, and the results interpreted in the light of the existing bibliography.

Response 12: In the revised manuscript, the bibliography is added at lines 253 and 257 to interpret our findings.

Line 251-253: “These vaccines affect the feelings of people as Raghupathi et al. demonstrate in their study that people are concerned about receiving newly developed vaccines”

Line 255-257 : “The sentiments related to each vaccine are mostly neutral and positive and in a study, Paul et al. found that only 14% of respondents had negative attitudes towards vaccines”.

Round 2

Reviewer 3 Report

We thank the authors for their efforts to include all the modifications and suggestions made. They have contributed to the improvement of the manuscript.